# Atomic-scale probing of short-range order and its impact on electrochemical properties in cation-disordered oxide cathodes

Linze Li[1,7], Bin Ouyang [2,3,7] ✉, Zhengyan Lun [2,4], Haoyan Huo [2], Dongchang Chen[5], Yuan Yue[5], Colin Ophus [6], Wei Tong [5], Guoying Chen [5], Gerbrand Ceder[2] ✉ & Chongmin Wang [1] ✉

Chemical short-range-order has been widely noticed to dictate the electro-chemical properties of Li-excess cation-disordered rocksalt oxides, a class of cathode based on earth abundant elements for next-generation high-energy-density batteries. Existence of short-range-order is normally evidenced by a diffused intensity pattern in reciprocal space, however, derivation of local atomic arrangements of short-range-order in real space is hardly possible. Here, by a combination of aberration-corrected scanning transmission elec-tron microscopy, electron diffraction, and cluster-expansion Monte Carlo simulations, we reveal the short-range-order is a convolution of three basic types: tetrahedron, octahedron, and cube. We discover that short-range-order directly correlates with Li percolation channels, which correspondingly affects Li transport behavior. We further demonstrate that short-range-order can be effectively manipulated by anion doping or post-synthesis thermal treatment, creating new avenues for tailoring the electrochemical properties. Our results provide fundamental insights for decoding the complex relationship between local chemical ordering and properties of crystalline compounds.

Chemical short-range-order (SRO) has been widely observed to govern material properties, such as strength and ductility of alloys[1–3], electro-chemical behavior[4–7], magnetic[8] and thermoelectric[9] characteristics. The presence of SRO in a bulk crystal can be routinely visualized in reciprocal space using diffraction techniques based on X-rays[10], neutrons[11], and electrons[12], where a diffused intensity pattern can be noticed. In electron diffractions, various two-dimensional (2D) diffused intensity pattern associated with different types of SRO clusters are often observed[13]. However, translating the diffused intensity pattern of reciprocal space to real space atomic configuration remains impos-sible, as the diffused intensity pattern only represents a compressed structural information averaged from a large region illuminated by the

beam of the diffraction techniques. Energy-filtered transmission elec-tron microscopy (TEM) allows direct visualization of SRO-related domain structures in real space, which is, however, limited to nanoscale resolution[2,14]. Electron tomography yields three-dimensional (3D) structural information of nanoscale objects[15–18], which has recently been applied to image metals at the atomic level in 3D[19–22]. However, atomic-level tomographic imaging of materials with multi-atomic species remains in its infancy[23]. The lack of a fundamental approach for direct atomic-level visualization of SRO in real space prevents direct correlation of atomic features of SRO with material properties.

Here, we develop a new framework based on the combination of aberration-corrected scanning transmission electron microscopy

[1]Environmental Molecular Sciences Laboratory, Pacific Northwest National Laboratory, Richland, WA, USA. [2]Department of Materials Science and Engi-neering, University of California – Berkeley, Berkeley, CA, USA. [3]Department of Chemistry and Biochemistry, Florida State University, Tallahassee, FL, USA. [4]School of Chemical Sciences, University of Chinese Academy of Sciences, Beijing, China. [5]Energy Storage and Distributed Resources Division, Lawrence Berkeley National Laboratory, Berkeley, CA, USA. [6]National Center for Electron Microscopy, Molecular Foundry, Lawrence Berkeley National Laboratory, Berkeley, CA, USA. [7]These authors contributed equally: Linze Li, Bin Ouyang. ✉e-mail: bouyang@fsu.edu; gceder@berkeley.edu; chongmin.wang@pnnl.gov

(STEM), electron diffraction, cluster-expansion Monte Carlo (CEMC) simulations, and simulations of STEM imaging and electron diffraction to directly correlate the SRO information in reciprocal space to local atomic-level cation arrangements in real space in one of the most promising materials for next-generation high-energy-density batteries: Li-excess cation-disordered rocksalt (DRX) oxides[6,24,25]. We show that, in the DRX lattice, any observable SRO pattern in reciprocal space is a convolution of three basic types of cation SRO: tetrahedron, octahedron, and cube. We directly map out the atomic-scale patterns of cation SRO in real space, discovering that these SRO patterns correlate with Li percolation channels and therefore critically affect the Li transport properties. We further demonstrate that the atomic-scale patterns of cation SRO can be tailored through anion doping or post-synthesis thermal treatment of the DRX oxides, enabling the direct modification of the electrochemical properties of DRX cathodes.

## Results

### Nature of SRO in the DRX lattice

We use a model system of a Mn-redox-based DRX oxide cathode, $Li_{1.2}Ti_{0.4}Mn_{0.4}O_{2.0}$ (LTMO), and a comparable oxyfluoride cathode, $Li_{1.2}Ti_{0.2}Mn_{0.6}O_{1.8}F_{0.2}$ (LTMOF) to delineate the nature of the cation distribution beyond the crystalline cubic rocksalt lattice. The electron diffraction patterns of LTMO and LTMOF are presented in Fig. 1a and b, respectively, where the sharp spots correspond to the Bragg diffraction of the cubic rocksalt lattice, and the diffused intensity patterns originate from the cation SRO. The diffused intensity patterns of LTMO are circle-like, while that of LTMOF is square-like, but with vanishing intensity at the corners of the square, indicating the different atomic arrangements of the SRO in LTMO and LTMOF.

To reveal the nature of the SRO that gives rise to the different diffused intensity patterns in the electron diffraction patterns of LTMO and LTMOF, we first establish the basic SRO structures. The cation SRO defines a short-ranged statistical correlation between the occupancy of different cations at neighboring sites in the lattice. In the FCC cation sublattice (Fig. 1c), there exist three different types of SRO, which correspond to three types of cation clusters, i.e., tetrahedron (Fig. 1d), octahedron (Fig. 1e), and cube (Fig. 1f). The tetrahedron-type SRO cluster includes correlation of only first nearest-neighbor (NN) cations; the octahedron-type cluster includes correlation of first/second NN cations; and the cube-type cluster includes correlation of second/third/fourth NN cations. Consequently, three basic types diffraction locus in reciprocal space can be derived as shown in Fig. 1d–f and the supplementary materials. The experimentally observed diffuse intensity patterns in LTMO and LTMOF are convolutions of these three basic types of diffraction locus in reciprocal space. To illustrate this concept, by using Fourier transformation of pair correlations, we calculate various diffraction locus with the coexistence of two among the three basic types of SRO in the FCC cation sublattice (Supplementary Figs. 1–3). We plot several calculated patterns in sequences following the transition from octahedron-type to cube-type SRO in Fig. 1g and from octahedron-type to tetrahedron-type SRO in Fig. 1h, respectively. Comparison of the calculated patterns (Fig. 1g, h) with the experimental ones (Fig. 1a, b) clearly indicates that the atomic structure of LTMOF presents an obviously larger fraction of cube-type SRO component, whereas that of LTMO is dominated by the octahedron-type SRO (Supplementary Fig. 4).

The above analysis is consistently corroborated by cluster-expansion Monte Carlo simulations (CEMC) by which the observed 2D structural features associated with SRO can be correlated to 3D atomic models. We calculate the electron diffraction patterns based on the CEMC-simulated structures of LTMO and LTMOF (Fig. 1i, j). Such electron diffraction patterns were obtained from an average of 1000 CEMC-simulated structures, and each of the structure models has dimensions of 2.5 nm × 2.5 nm × 2.1 nm. Therefore, the average diffraction pattern among 1000 structures would effectively reflect the electron diffraction information at a particle with a length scale of 25 nm × 25 nm × 21 nm. Consistent with the experimental observations, the simulated electron diffraction pattern of LTMO (Fig. 1i) is dominated by the octahedron-type SRO feature (circle-like diffused intensity patterns), whereas in the simulated electron diffraction pattern of

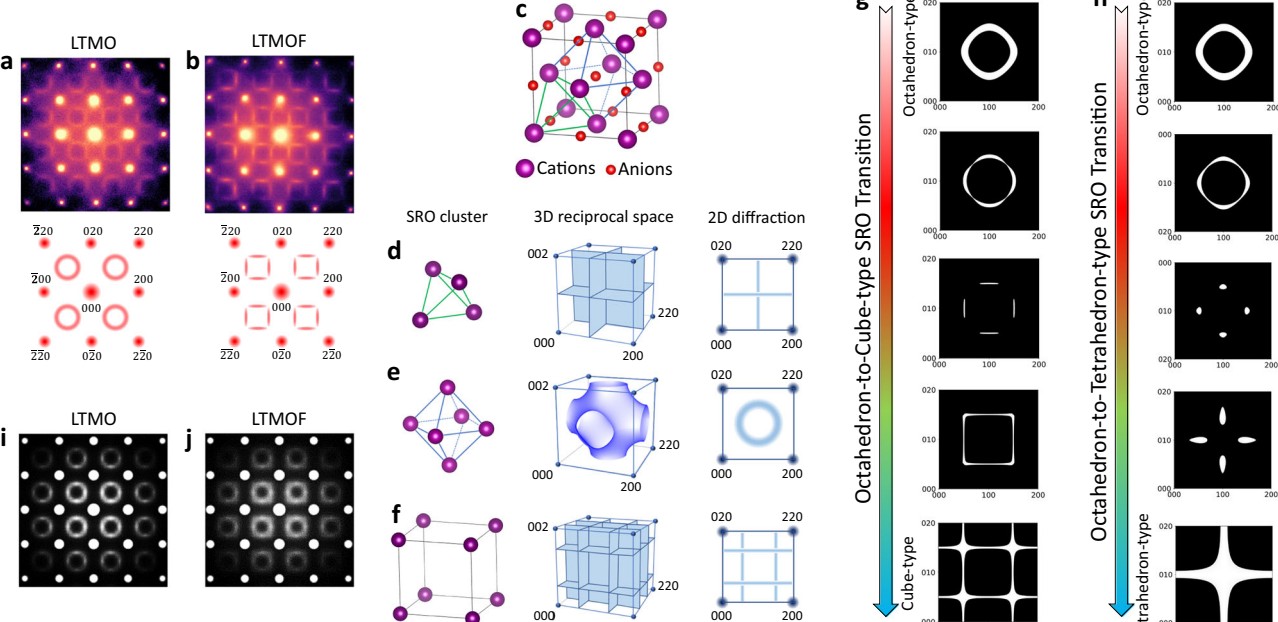

**Fig. 1 | The nature of SRO patterns in DRX. a**, **b** Experimentally observed electron diffraction patterns (top) along [001] zone axis of LTMO **a** and LTMOF **b**, and their corresponding schematics (bottom). **c** Crystal structure of DRX, in which different atomic clusters are highlighted by the colored polyhedrons. **d–f** Octahedron-type SRO cluster **d**, cubic-type SRO cluster **e**, and tetrahedron-type SRO cluster **f** (left), and the corresponding calculated diffraction locus in reciprocal space (middle) and diffraction patterns along [001] zone axis (right). **g**, **h** Calculated diffraction locus for the octahedral-to-cubic SRO transition **g** and octahedral-to-tetrahedral SRO transition **h**. **i**, **j** Simulated electron diffraction patterns for LTMO **i** and LTMOF **j**.

LTMOF (Fig. 1j), an enhanced feature of cube-type SRO (square-like patterns) is observed.

It is worth mentioning that our analysis of the three types of SRO features is based on our observation that these features frequently show up during both experiments and simulations. Our analysis does not eliminate the existence of other possible SRO feature; however, we also did not see other important SRO feature in the material systems we investigated. Moreover, lower-dimension ordering within local clusters, such as 1D dimer, 2D triangle or square may also exist, since they are the building blocks of the 3D tetrahedron, octahedron, and cube type SRO. Our analysis implies that the lower-dimension local ordering, if existing, has to satisfy the constraint as defined by the 3D SRO structures.

### Direct mapping of Li percolation channels and their correlation with SRO

To reveal the atomic-scale structural features of SRO and correlate them with Li percolation in the DRX cathodes, here we develop a high-angle annular dark-field (HAADF) STEM-imaging (Z-contrast image) method to directly extract the spatial correlation of Li-rich and TM-rich channels in the DRX lattice. By mapping out the intensity distribution of the atomic columns in the Z-contrast image, the configuration of Li-rich or TM-rich channels at the image plane can be directly revealed

(Supplementary Figs. 5 and 6). To illustrate this, we use a nanosized 3D supercell of 4.2 nm in thickness that is derived from CEMC simulation to calculate the Z-contrast image (Fig. 2a–f). For the 3D supercell, the number of Li atoms within each atomic column can be counted (Fig. 2a, b), where the Li-rich and TM-rich (thus Li-poor) atomic columns can be illustrated by a color map of 2D projection (Fig. 2c). The nanoscale patterns in the intensity map of the calculated Z-contrast image (Fig. 2f) matches with the map of cation distributions (Fig. 2c), demonstrating that we can use intensity map of Z-contrast image to directly derive cation distribution. It should be noticed that, as detailed in the supplementary materials, for the experimentally captured Z-contrast images, the depth of field is 5.9 nm, which is comparable with thickness of 4.2 nm in the supercell.

The atomic structures and local ordering patterns of LTMO are revealed by the Z-contrast image (1st panel in Fig. 2g). Circle-like diffused intensity patterns associated with the octahedron-type SRO are observed in the fast Fourier transform (FFT) of the Z-contrast image (2nd panel in Fig. 2g), which is consistent with the electron diffraction result in Fig. 1a. The Z-contrast-image intensity color map (3rd panel in Fig. 2g) reveals the atomic-scale configurations of the Li-rich (green) and TM-rich (red) nanoregions, showing both types of nanoregions are atomically thin. To reveal the connectivity of Li-rich nanoregions, the intensity map is plotted with a threshold to show only the Li-rich

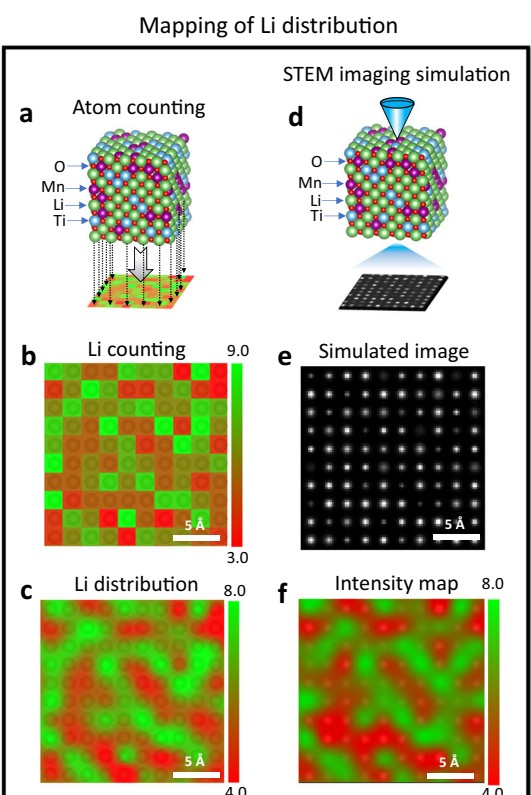

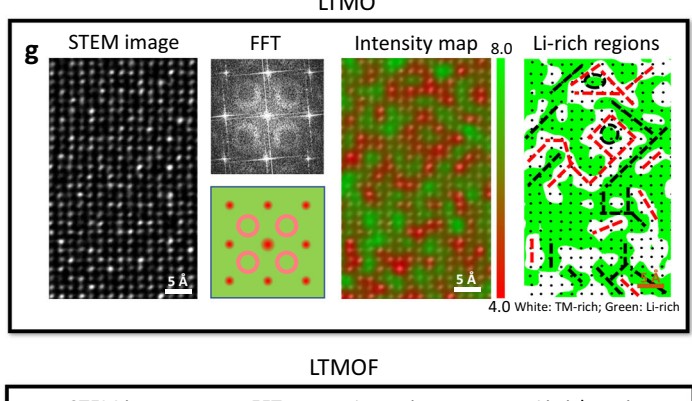

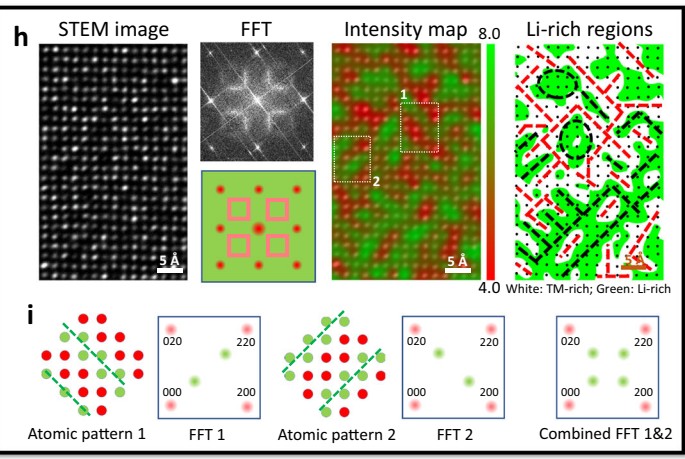

**Fig. 2 | Correlation of SRO patterns with Li transport properties. a** Nanosized 3D supercell structure calculated from the Monte Carlo simulations. The number of different atoms within each atomic column in the 3D supercell structure can be counted. **b, c** Pixel map **b** and corresponding contour color map **c** for the counted numbers of Li atoms in different atomic columns. In both maps, green and red correspond to Li-rich and TM-rich (thus Li-poor) atomic columns, respectively. **d** A Z-contrast image can be simulated from the 3D supercell structure. **e, f** Simulated Z-contrast image **e** and corresponding intensity map **f** for the atomic columns in the Z-contrast image. In the intensity map, green and red correspond to lower and higher intensities, respectively. **g** Atomic-scale STEM Z-contrast image of LTMO

(leftmost) and its corresponding fast Fourier transform (FFT) (second left), intensity color map showing the distribution of both Li-rich (green) and TM-rich (red) nanoregions (third left), and the same intensity map showing only the Li-rich (green) nanoregions (rightmost). **h** Atomic-scale STEM Z-contrast image of LTMOF (leftmost) and corresponding FFT (second left) and intensity color maps showing the distribution of both Li-rich (green) and TM-rich (red) nanoregions (third left) and showing only the Li-rich (green) nanoregions (rightmost). **i** Atomic models for the highlighted regions in the intensity map in **h** and corresponding schematic FFT patterns.

(green) nanoregions (the 4th panel in Fig. 2g), indicating that most of the Li-rich nanoregions are connected to each other, while isolated Li-rich nanoregions can be seen (such as those highlighted by the dashed circles), which is enclosed by atomically thin TM-rich regions.

For LTMOF, the diffuse intensity is featured by the square-like patterns, indicating the dominance of cube-type SRO (2nd panel in Fig. 2h), which is consistent with the electron diffraction result as shown in Fig. 1b. The intensity map of Z-contrast image of LTMOF (3rd panel in Fig. 2h) shows enhancement of local ordering of atomically thin Li-rich (green) and TM-rich (red) nanoregions. A strong tendency of forming alternating Li-rich and TM-rich layers oriented along [110] or [1-10] directions (i.e., the diagonal direction) of the cubic cell can be noticed, which is highlighted by the dashed rectangles in the intensity map and correspondingly illustrated by the schematics in Fig. 2i. The observed layered-like patterns differ from the local feature of conventional layered materials, where alternating Li and TM layers are present in the [111] direction of a cubic cell, but can be described by an unconventional layered ordering in analogy to AgZr-type metallic alloys[26] (Supplementary Fig. 7). Such unconventional-layered-like patterns are frequently observed within local nanosized domains with width of several unit cells. Because of the overall disordered nature at the cation sites, two perpendicular unconventional-layered-like patterns, pointing along either the [110] or [1-10] direction, have equal probabilities to exist in LTMOF, which results in the formation of interconnected Li-rich stripes, as well as cage-like structures of TM-rich stripes with enclosed Li-rich nanoregions (such as those highlighted by the dashed circles in the rightmost panel in Fig. 2h). The isolated Li-rich nanoregions in LTMOF appear to be larger than those in LTMO, indicating the connectivity of Li-rich networks in LTMO is better than that in LTMOF.

The experimental observations in Fig. 2g–i clearly indicate that the formation of cube-type SRO in LTMOF leads to a tendency of the ordered pattern of cation distribution within localized nanoscale domains, whereas the octahedron-type SRO-dominated LTMO results in a tendency of disordered cation distribution. Our simulation framework provides insights for such enhanced localized cation ordering in LTMOF. We evaluate the order-disorder phase transition temperature by sampling the specific heat capacity ($C$) as a function of temperature ($T$) (Fig. 3a) as detailed in the method section. In principle, order-disorder phase transformation will occur with increasing temperature, and the heat capacity will peak at the phase-transition point, while the change of heat capacity after the peak can be correlated to the evolution of SRO[27]. The calculated $C-T$ curves thus indicate a higher transition temperature of LTMOF relative to LTMO, which also suggests that at the same synthetic temperature (1273 K), LTMOF should have stronger local ordering than LTMO.

To examine the effects of SRO on the Li transport properties, we performed Li percolation analysis in the simulated LTMO and LTMOF structures, where the dominating octahedron-type SRO in LTMO (Fig. 1i) and an enhanced cube-type SRO in LTMOF (Fig. 1j) were reproduced in the simulation. Such analysis is based on calculating the 3D networks that are composed of Li-pure (0-TM) tetrahedrons in the DRX lattice[6,28,29]. The analysis (Fig. 3b) shows that the LTMO and LTMOF compounds have 0.326 and 0.226 Li percolating within the bulk lattices, respectively, which indeed confirms the better connectivity of Li-rich networks in LTMO and is thus consistent with the experimental observation for the 2D projection (Fig. 2g–i). Additionally, we also performed percolation analyses on several long-range-ordered rocksalt-based structures (Fig. 3b). Both LTMO and LTMOF showed less 0-TM accessible Li compared with the well-known layered structure, the unconventional layered structure (defined in Supplementary Fig. 7), and the DRX structure without SRO (i.e., with a fully random cation distribution) but much more 0-TM accessible Li than the γ-LiFeO$_2$ structures.

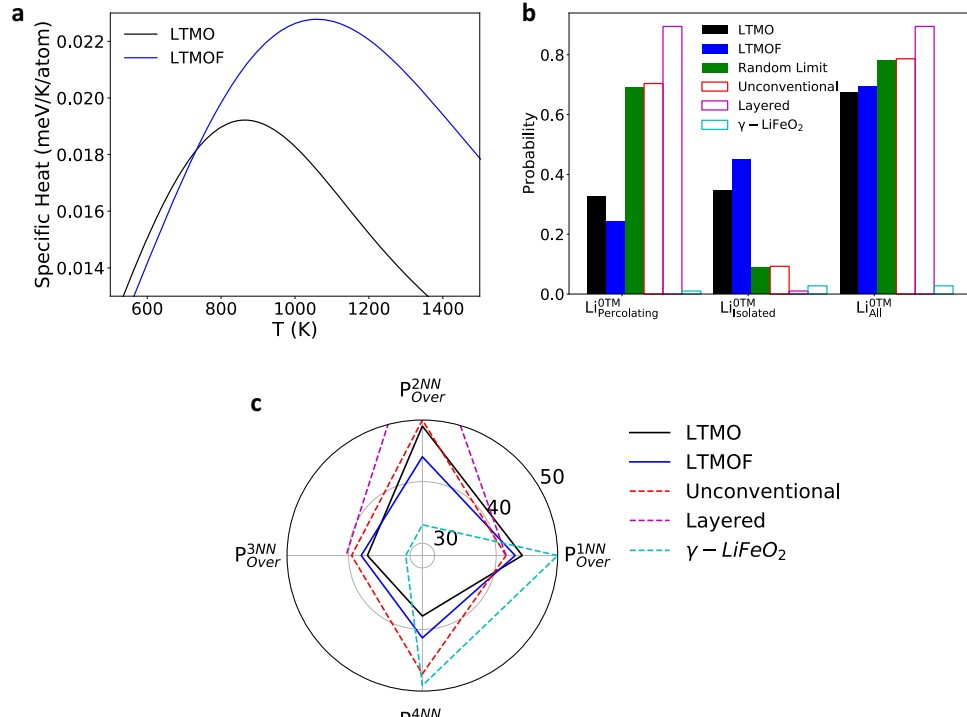

**Fig. 3 | Theoretical calculation of DRX and related structures. a** Calculated specific heat evolution between 500 and 1500 K for LTMO and LTMOF. **b** 0-TM percolating properties for LTMO and LTMOF, the DRX structure with a fully random cation distribution, and three long-range-ordered structures (unconventional layered, layered, and γ-LiFeO$_2$ type) with the same Li contents. **c** Probability of finding over-coordinated Li$^+$ in LTMO and LTMOF and their comparisons with the DRX structure with a fully random cation distribution and the three long-range-ordered structures with the same Li contents. The labels on the radial axis have units of percentage.

To gain insight on how the connectivity of Li-rich networks evolves from LTMO to LTMOF, we calculated the over-coordination probability in different coordination shells of Li (Fig. 3c). Such over-coordination probability is defined as the probability of finding more TM in each coordination shell of Li in LTMO or LTMOF than in a structure with a fully random cation distribution. A larger over-coordination probability indicates a stronger correlation between a Li ion and its neighboring TMs. Compared with the case of LTMO, a Li ion in LTMOF tends to have less TMs at its 1st nearest neighboring (1NN) and 2NN sites (smaller 1NN and 2NN over-coordination probabilities) and more TMs at its 3NN and 4NN sites (larger 3NN and 4NN over-coordination probabilities). The smaller 1NN and 2NN over-coordination probabilities in LTMOF lead to more 0-TM channels, whereas the larger 3NN and 4NN over-coordination probabilities result in the interruption of the connectivity of 0-TM channels, which could be at the origin of the more isolated Li-rich regions observed in LTMOF. It is also worth mentioning that when the 1NN over-coordination probability is small enough, the total amount of 0-TM channels will become sufficient such that the Li-rich networks will always be well connected. This is evident in the DRX structure with fully random cation distribution as well as in the well-ordered layered and unconventional layered structures (Fig. 3b, c). In contrast, the exceptionally large 1NN over-coordination probability in γ-LiFeO₂ leads to almost no 0-TM channels and therefore no 0-TM accessible Li.

## Tailoring SRO to tune electrochemical properties of DRX cathodes

The insights on the correlation between SRO and Li transport properties provide general guidance for modifying the electrochemical properties of DRX cathodes by tuning the SRO features. The observed distinguished differences in the structural features of LTMO and LTMOF clearly demonstrate the effect of fluorination on the SRO (Fig. 1a, b) and the associated atomic-scale patterns of Li-rich percolating networks (Fig. 2g–i). The effect of F doping on the electrochemical performance is clearly reflected in the voltage profiles in Fig. 4a, where the 1st-cycle charge and discharge capacities measured in the LTMO cathode are higher than those measured in the LTMOF cathode. Given that the two DRX cathodes have the same long-range chemical order and similar particle size and morphology, and that no obvious structural degradation was observed at the surfaces of either cathode (Supplementary Figs. 8–13 and Supplementary Table 1), it is concluded that the enhanced 1st-cycle capacities of LTMO are mainly caused by the improved connectivity of the Li-rich percolating channels regulated by the local SRO (Fig. 2).

We expect that the Li connectivity can be improved by enhancing local unconventional layered ordering, i.e., by increasing the size of the unconventional-layered-like domains in the DRX lattice. We demonstrate that such enlargement of unconventional-layered-like domains associated with the cube-type SRO in the DRX lattice can indeed be

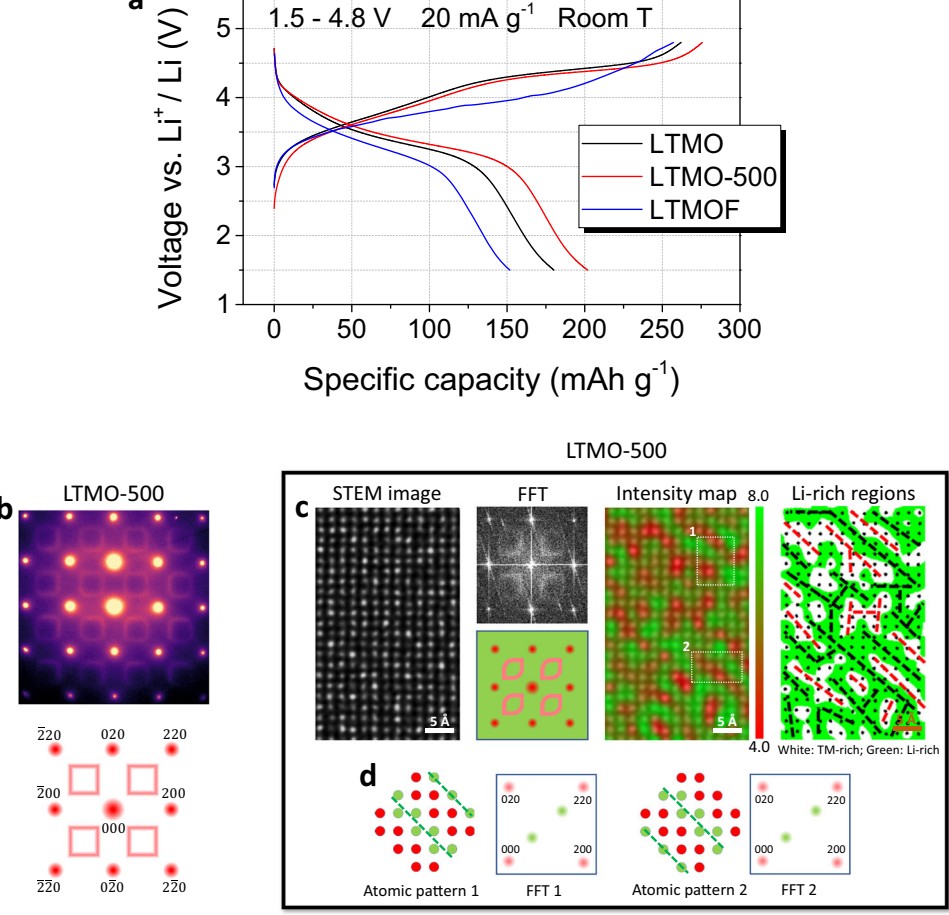

**Fig. 4 | Tailoring electrochemical properties of DRX cathodes by tuning SRO. a** Charge and discharge voltage profiles for the 1st cycle of LTMO, LTMOF, and LTMO-500 (LTMO annealed at 500 °C for 60 h) cathodes in half-cells cycled between 1.5 and 4.8 V at a current density of 20 mAg⁻¹. **b** Electron diffraction pattern (top) of LTMO-500 and the corresponding schematic (bottom). **c** Atomic-scale STEM Z-contrast image of LTMO-500 (leftmost) and corresponding FFT (second left) and intensity color maps showing the distribution of both Li-rich (green) and TM-rich (red) nanoregions (third left) and showing only the Li-rich (green) nanoregions (rightmost). **d** Atomic models for the highlighted regions in the intensity map in (**c**) and corresponding schematic FFT patterns.

achieved by annealing the LTMO cathodes at 500 °C for 60 h. The annealing process does not alter the long-range chemical order or particle morphology of the annealed LTMO (LTMO-500) particles as detailed in Supplementary Figs. 8–13 and Supplementary Table 1, but only modifies their SRO features. In the electron diffraction pattern of LTMO-500 (Fig. 4b), full square-like diffraction patterns are observed, indicating dominance of cube-type SRO. The square-like diffraction patterns of LTMO-500 is featured by non-vanishing intensity at the square corners, which differs from those of LTMOF. In the FFT of the *Z*-contrast image of LTMO-500, square-like patterns are also observed; however, in each square, two of the corners are surprisingly rounded (2$^{nd}$ panel in Fig. 4c). This asymmetric shape in the FFT indicates the dominance of only one type of quasi-periodic patterns in the local nanoscale area. As shown in the *Z*-contrast-image intensity map (3$^{rd}$ panel in Fig. 4c) and corresponding schematics (Fig. 4d), the atomically thin Li-rich and TM-rich stripes are predominantly oriented along the diagonal pointing from the top left to bottom right, leading to a well-connected Li-rich network (4$^{th}$ panel in Fig. 4c). The well-connected Li-rich networks have been repetitively observed at different local areas (Supplementary Fig. 14), and consequently, LTMO-500 delivered higher charge and discharge capacities than LTMO and LTMOF (Fig. 4a).

## Discussions

The consistence between theoretical prediction and electrochemical performance validates that theoretical framework of 0TM percolation is useful for understanding the design handle for promoting Li diffusion. However, it is worth mentioning that such analysis is only based on the cation distribution of fully lithiated DRX structures, while assuming the formation of 0TM local structure. In that case, such framework does not consider the impact of other local structures and the possibilities of large-scale rearrangement of cations after cycling. It should be noted that SRO may also affect electron transfer and consequentially anionic redox potential. In DRX, cation-disordering allows the formation of a significant amount of linear "Li-O-Li" configuration which produces states that can be oxidized[30], and is therefore regarded as the key structural origin of oxygen redox. With a high Li-excess content of 20% in the model materials, there should already be more than enough "activated" O centers with Li-O-Li configuration. Moreover, it has been demonstrated that the Li transport kinetics, instead of the amount of (anionic) redox reservoir, is the critical control of the accessible capacity of DRX materials[7,29,31,32]. It would be expected that, as comparing LTMO with LTMO-500, the anionic-redox will not show significant difference, especially considering these two materials have the exact same composition.

In conclusion, cation SRO in DRX has been previously noticed as complex diffused intensity patterns in reciprocal space. We have developed a framework to deconvolute the compressed information into fundamental atomic-level structures and correlated these atomic-scale SRO structures to Li transport properties in DRX cathodes. It is further demonstrated that the cation SRO features can be effectively manipulated by anion doping or post-synthesis thermal treatment, correspondingly leading to the modification of electrochemical performance of DRX cathodes. The approach we have developed here can be generally used for direct atomic-scale mapping of chemical SRO features in a crystalline solid solution involving multi-cation and multi-anion species.

## Methods
### Material synthesis
The cathodes designated as $Li_{1.2}Ti_{0.4}Mn_{0.4}O_{2.0}$ (LTMO) and $Li_{1.2}Ti_{0.2}Mn_{0.6}O_{1.8}F_{0.2}$ (LTMOF) were prepared through a conventional solid-state reaction process. Starting materials such as $Li_2CO_3$ (sourced from Alfa Aesar with an ACS purity of at least 99%), $Mn_2O_3$ (with a purity of 99.9% from Alfa Aesar), $TiO_2$ (99.9% pure, Alfa Aesar), and LiF

(Alfa Aesar, with a purity of 99.99%) served as the precursor substances. These precursors were proportionally blended in ethanol (factoring in an extra 10% of $Li_2CO_3$ and LiF to make up for potential losses during the synthesis process) and milled using a Retsch PM 400 planetary ball mill at a speed of 180 revolutions per minute for a duration of 12 h. After mixing, the materials were dried overnight at 70 °C and formed into pellets. These pellets were subjected to a sintering process at a temperature of 1000 °C in an argon-filled environment for four hours. Post-sintering, the pellets were rapidly cooled in an argon atmosphere, moved to a glovebox, and then pulverized into fine powders.

### Electrochemical test
The cathode films were fabricated using a mix of active material, Super C65 carbon black (Timcal), and polytetrafluoroethylene (PTFE, sourced from DuPont under the Teflon 8 A brand) in a weight ratio of 70:20:10. To construct these films, 350 mg of the freshly produced active material and 100 mg of Super C65 were combined and subjected to ball milling at a speed of 250 rotations per minute for 6 h under an argon shield. This process was performed using a Retsch PM200 planetary ball mill. Subsequently, PTFE was incorporated into the blend by hand-stirring with the previously milled mixture for an additional 40 min. The mixture was then processed into thin film form within the confines of a glovebox. For the electrolyte, a commercial solution of 1 M $LiPF_6$ dissolved in a mixture of ethylene carbonate (EC) and dimethyl carbonate (DMC) at an equal volume ratio was utilized (procured from Sigma). A glass microfiber filter from Whatman served as the separator between the components. Lithium metal foil from FMC was employed as the anode material. The assembly of coin cells was carried out in a glovebox, and their performance was evaluated using an Arbin battery testing system at ambient temperature. The films were characterized by a loading density ranging from approximately 3 to 4 mg cm$^{-2}$, based on the weight of the active materials involved.

### Scanning/Transmission electron microscopy (S/TEM)
The LTMO and LTMOF cathode samples intended for transmission electron microscopy (TEM) were carefully placed onto TEM lacey carbon grids within an argon-saturated glovebox to prevent exposure to any solutions during the preparation for TEM analysis. The morphology and diffraction studies were conducted using a Titan 80-300 scanning/transmission electron microscope (S/TEM) equipped with aberration correction and set to a voltage of 300 kV. Similarly, aberration-corrected scanning transmission electron microscopy (STEM) was carried out on a JEOL JEM-ARM200CF microscope, utilizing a 200 kV operational voltage. The convergence semi-angle was fixed at 27 milliradians. For high-angle annular dark-field (HAADF) STEM imaging, signals were collected across semi-angles ranging from 68 to 280 milliradians. With the given experimental parameters, the depth of field was calculated to be approximately 5.9 nm (calculated as approximately 1.77 times the electron beam wavelength, λ, which is 2.51 picometers at 200 keV, divided by the square of the convergence semi-angle, α, 27 mrad). For the STEM Z-contrast imagery, processing involved the application of a two-dimensional Gaussian function to each bright-contrast dot, which corresponds to an atomic column in the imagery. The intensity map was then produced based on the height of the Gaussian fit at each of these dots, as illustrated in Supplementary Fig. 11.

### Energy-dispersive X-ray spectroscopy (EDS)
Energy-dispersive X-ray spectroscopy (EDS) data were acquired using a Titan 80–300™ scanning/transmission electron microscope (S/TEM) with aberration correction, operated at a 300 kV voltage, and equipped with an Oxford X-Max TEM EDS detector. The subsequent analysis of the EDS data was conducted with the "AZtec" software suite. This

software facilitates the separation of overlapping spectral peaks by referencing pre-recorded standard spectra. A Filtered Least Squares (FLS) method is applied within the software to accurately delineate the signal peaks from the background noise.

### Electron energy loss spectroscopy (EELS)

Electron energy loss spectroscopy (EELS) microanalysis was performed on the aberration-corrected JEOL JEM-ARM200CF microscope operating at a voltage of 200 kV, paired with a Gatan Image Filter (GIF) that functions with an energy dispersion of 0.25 eV per channel. The electron probe was configured with a convergence semi-angle of 20.6 milliradians, and the EELS data were gathered with a collection semi-angle of 42.9 milliradians. Processing of the EELS spectra involved the removal of the pre-edge background, which follows a power-law distribution, using the Digital Micrograph software.

### Density functional theory calculations and cluster expansion construction

First-principles density functional theory (DFT) calculations were performed to fit the cluster expansion models that was later used for Monte Carlo sampling of different states of chemical short-range order. All the calculations employed the projector-augmented wave (PAW) method[33] as implemented in the Vienna Ab initio Simulation Package (VASP version 5.4.2)[34]. The rotationally averaged Hubbard U correction[35,36] was used to correct the self-interaction error in Mn. The U parameter was obtained from a earlier reported calibration to oxide formation energies[36]. For all the calculations, a reciprocal space discretization of 25 k-points per $Å^{-1}$ was applied, and the convergence criteria were set as $10^{-6}$ eV for electronic loops and 0.02 eV $Å^{-1}$ for ionic loops.

A cluster expansion model was subsequently trained by DFT calculations to model the complete configurational and compositional space of $Li^+$- $Mn^{3+}$- $Ti^{4+}$- $O^{2-}$-$F^-$. In the cluster expansion, the configurational energy dependence is captured by an expansion into different cluster functions, which can be expressed as[37,38]:

$$E = \sum_{i,sp1} J_i^{sp1} \sigma_i^{sp1} + \sum_{i,j,sp1,sp2} J_{ij}^{sp1sp2} \sigma_i^{sp1} \sigma_j^{sp2} + \sum_{i,j,k,sp1,sp2,sp3} J_{ijk}^{sp1sp2sp3} \sigma_i^{sp1} \sigma_j^{sp2} \sigma_k^{sp3} \cdots\cdots$$
$$(1)$$

Here, $\sigma_i^{sp}$ represents the occupancy of a certain site(s) with a certain species $sp$ and $J$ corresponds to the effective cluster interactions (ECIs). To construct the cluster expansion, pair interactions up to 7.1 Å, triplet interactions up to 4.0 Å, and quadruplet interactions up to 4.0 Å based on a rocksalt lattice with a cubic lattice parameter of $a = 3.0$ Å were used in the cluster-expansion formulism. The ECIs were fitted to density-functional theory (DFT) energies of sampled structures using a L1-regularized least-squares regression approach[39], with the regularization parameters chosen to minimize cross-validation error[29,39–42]. The DFT results of 1251 structures were applied to fit the cluster expansion, which ended up with a cluster expansion model with root-mean-squared error of 7.5 meV/atom.

### Monte Carlo sampling and free energy calculations

With the constructed cluster expansion, we then performed Monte Carlo simulation at 1273 K for both LTMO and LTMOF. For each composition, electron diffraction simulations, percolation analysis, and SRO analysis were performed on 10000 MC structures with 1440 atom cells[28,29,43]. All the simulations were first equilibrated for eight million steps accompanied by another eight million steps of production run.

Similarly, cluster expansion Monte Carlo simulation was also used for calculating the internal energy as a function of temperature from canonical ensemble. In this case, a 960-atom supercell was used to sample the internal energy. Each simulation consists of ten million of equilibration run and ten million of production run. With the sampled free energy, the temperature-dependent entropy was calculated by

integrating the calculated heat capacity from the infinite random limit, which can be indicated by the following equation:

$$S = k_B S_{Ideal} + \int_{\infty}^{T} \frac{C_V}{T} dT \qquad (2)$$

### Characterization of cathode particles

Scanning electron microscopy (SEM) images and X-ray diffraction (XRD) data of the samples of LTMO, LTMOF, and LTMO-500 cathode particles are presented in Supplementary Fig. 5. TEM images of the three samples are shown in Supplementary Fig. 6. As observed in the SEM images and TEM images, the particles in all three samples exhibit similar morphology and size, mostly 1-5 um in diameter, while smaller sub-micron particles can also be observed. The XRD data confirms the cubic rocksalt structure with *Fm-3m* symmetry with no observable impurity peaks in all three samples. The lattice constants of LTMO, LTMOF, and LTMO-500 are 4.1458 Å, 4.1540 Å, and 4.1496 Å, respectively. The STEM images and corresponding energy-dispersive X-ray spectroscopy (EDS) elemental maps in Supplementary Fig. 7 show that the cations and anions are uniformly distributed at the particle level in all three samples. The EDS-measured atomic percentage compositions averaged in the particle bulk of the three samples are shown in Supplementary Table 1.

To explore any chemical redistribution or changes of the oxidation states near the surfaces of LTMO, LTMOF, and LTMO-500 cathode particles, EELS elemental maps and EELS spectra of the O K-edges and Ti and Mn L-edges collected at 7 different locations below the particle surfaces of the three samples are presented in Supplementary Figs. 8-10. Note that due to the low concentration of F in LTMOF, the F K-edges cannot be detected by EELS in our work. In general, the EELS elemental maps show that Ti, Mn, and O are uniformly distributed near the surfaces of all three samples. And reduction of Ti and Mn valences, as evidenced by the suppression of the Ti $L_2$ and $L_3$ pre-peak intensities and shift of the Mn $L_3$ peak to lower energies[44,45], is observed within the very surface layers (~5-nm thick) of all three samples.

### Derivation of diffraction locus in the reciprocal space for tetrahedron-type, octahedron-type, and cube-type SRO

To understand the origins for the observed electron diffraction patterns of LTMO and LTMOF, it is necessary to identify the local SRO structures (or atom clusters) that are responsible for certain diffraction locus in the reciprocal space. In the FCC lattice of DRX, three fundamental local SRO structures, i.e., and tetrahedron-type cluster (Fig. 1f), octahedron-type cluster (Fig. 1d), and cube-type cluster (Fig. 1e), can exist. The cation SRO defines a short-ranged statistical correlation existing between the occupancy of different cations at neighboring sites in the lattice. The tetrahedron-type SRO cluster includes correlation of only first nearest-neighbor (NN) cations; the octahedron-type cluster includes correlation of first/second NN cations; and the cube-type cluster includes correlation of second/third/fourth NN cations. The cluster-type SRO indicates that the composition within the cation clusters tend to conserve or approach the bulk composition. As a result, the probability of finding a Li-TM pair between one cation and its first NN cations (tetrahedron-type), or its first and second NN cations (octahedron-type), or its second, third, and fourth NN cations (cube-type) should conserve a linear relationship[13,43]. More specifically, the relationship is:

$$1 + 3\alpha_1 = 0 \text{(tetrahedron type)} \qquad (3)$$

$$1 + 4\alpha_1 + \alpha_2 = 0 \text{(octahedron type)} \qquad (4)$$

$$1 + 3\alpha_2 + 3\alpha_3 + \alpha_4 = 0 \text{(cubic type)} \qquad (5)$$

in which $\alpha_i$ is the Warren-Cowley SRO parameter at the $i^{th}$ NN. According to the definition of Warren-Cowley SRO parameter, $\alpha_i$ can be calculated by inverse Fourier transformation of the diffraction intensity caused by a SRO, which can be formulated as:

$$\alpha_i = \int_{V^*} I_D(g) \exp[2\pi i \mathbf{r}_i g] dg \tag{6}$$

where $I_D$ refers to the diffraction intensity caused by the SRO at the $i^{th}$ NN coordination shell. $\mathbf{r}_i$ refers to the pair vector at the $i^{th}$ NN coordination shell, $g$ refers to the base vector at reciprocal space. By substituting (6) into (3), (4), and (5) for a specific SRO, we get:

$$\int_{V^*} I_D(g) F(g) \, dg = 0 \tag{7}$$

In Eq. (7), $F(g)$ is formulated as:

$$F(g) = \sum_i \exp[2\pi i g \mathbf{r}_i] \tag{8}$$

where the summation is taken within all the $i^{th}$ pair interactions among a certain SRO cluster. As being indicated in Eq. (7), when $I_D(g) \neq 0$ occurs, we should have $F(g) = 0$, while the solution on reciprocal space (g) would be the corresponding SRO diffraction locus as shown in Fig. 1[13]. For octahedron-type, cube-type, and tetrahedron-type SRO, the geometric locus should satisfy:

$$\begin{cases} \cos\left(\frac{\pi h}{2}\right) \cos\left(\frac{\pi k}{2}\right) \cos\left(\frac{\pi l}{2}\right) = 0 \\ \sin\left(\frac{\pi h}{2}\right) \sin\left(\frac{\pi k}{2}\right) \sin\left(\frac{\pi l}{2}\right) = 0 \end{cases} \text{(tetrahedron type)} \tag{9}$$

$$\cos \pi h + \cos \pi k + \cos \pi l = 0 \,\text{(octahedron type)} \tag{10}$$

$$\cos \pi h \cos \pi k \cos \pi l = 0 \,\text{(cube type)} \tag{11}$$

**Qualitative simulation of the mixture of different types of SRO**
With the derived diffraction locus above, we can simulate different conditions for a mixing of the three fundamental types of SRO, i.e., tetrahedron-type, octahedron-type, and cube-type. As indicated in Eq. (7), the diffraction intensity will appear in the reciprocal space where $F(g) = 0$. However, in real electron diffraction experiments, the diffraction locus would always follow $F(g) = \delta$, where $\delta$ represents a distribution of values near zero. The non-zero values of $\delta$ for real experiments can be caused by imperfection of electron optics in TEM instruments, local non-stoichiometric composition in the materials, and so on[13]. For the simulations of Fig. 1, we approximate the $\delta$ with a Gaussian function with the formula:

$$\delta = e^{-\frac{1}{\sqrt{2\pi}\sigma}\left(\frac{|F(g)|}{2\sigma}\right)^2} \tag{12}$$

With such definition, larger $\sigma$ would lead to more centralized intensity, which corresponds to stronger SRO features. We can tune the values of $\sigma$ for $F(g)$ for different types of SRO to simulate the cases when two types of SRO are mixed. Schematic of different mixing conditions of octahedron-type, cube-type, and tetrahedron-type SRO are demonstrated in Supplementary Figs. 1–3.

## Data availability
All data that support the findings of this study have been included in the main article and Supplementary Information or from the corresponding authors upon request. Source data are provided in this paper.

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

## Acknowledgements

This work was supported by the Assistant Secretary for Energy Efficiency and Renewable Energy (EERE), Vehicle Technologies Office (VTO), of the U.S. Department of Energy (DOE) under the Cation-disordered Rocksalt Cathode Materials (DRX + ) consortium (DE-LC-000L096 to C.W. and G.C.). Part of the work was conducted at the William R. Wiley Environmental Molecular Sciences Laboratory (EMSL), a national scientific user facility sponsored by DOE's Office of Biological and Environmental Research and located at Pacific Northwest National Laboratory (PNNL). PNNL is operated by Battelle for the Department of Energy under Contract DE-AC05-76RL01830.

## Author contributions

L.L. carried out the STEM analysis, B.O. carried out the CEMC simulation, B.O. also carried out the STEM image simulation with guidance from C.O., Z.L., H.H., D.C., Y.Y., W.T., and G.Y.C. synthesized the samples and carried out the electrochemical testing, G.C. and C.W. supervised the research. L.L. and B.O. analyzed the data and drafted the manuscript with final approval by all authors.

## Competing interests

The authors declare no competing interests.
