## [Peer Review File · Nature Communications]

Atomic-scale probing of short-range order and its impact on electrochemical properties in cation-disordered oxide cathodesREVIEWER COMMENTS

Reviewer #1 (Remarks to the Author):

The manuscript provides a way to unravel short-range order (SRO) in Li-excess cation-disordered rocksalt (DRO) using aberration-corrected scanning transmission electron microscopy, electron diffraction, and cluster-expansion Monte Carlo simulations. The authors also claim that they can tune the SRO by substitution of transition metals.

I have the following questions:

- 1) If the SRO in DRX are a convolution of octahedron, tetrahedron and cube type, is this a dynamic or a static ordering?
- 2) How do the atomic-scale STEM Z-contrast images and the fast Fourier transform (FFT) intensity color maps correlate with diffusion properties determined with e.g. GITT or impedance spectroscopy.
- 3) Why are you using a coating of 70:20:10 and how do you ensure that diffusion properties are not dominated by the carbon/binder matrix?

Reviewer #2 (Remarks to the Author):

This study combines first-principles calculations and experiments to address a significant challenge: the direct atomistic-level visualization of chemical short-range order (SRO) in bulk oxide crystals. Visualizing and, therefore, better understanding SRO is crucial for enhancing the functionality and design of materials across various applications. To tackle this challenge, the authors employ aberration-corrected STEM, electron diffraction, and first-principles simulations, as well as STEM image and electron diffraction simulations. They specifically focus on two Li-excess cation-disordered rocksalt oxides, namely LTMO and LTMOF, as case studies, which is sound. The authors also demonstrate how SRO features can be tailored through anion substitution and thermal treatment, enabling the modification of the electrochemical performance of these materials.

The research findings are presented in a clear and concise manner, and the conclusions drawn seem robust, offering valuable insights for characterizing and designing new electroactive materials. Moreover, experimentalists interested in synthesizing the proposed compounds can benefit from this study. I believe this work holds significant scientific and technological interest, and the methods and approaches described here can be extended to explore other chemical spaces and materials, thus broadening the impact of the research within the field. Overall, the study is comprehensive and well-written. In my opinion, with a few minor revisions as noted below, the manuscript can be published in Nature Communications:

1. One aspect that requires clarification is the limited consideration of SRO types restricted to tetrahedron, octahedron, and cube clusters. While these clusters are indeed significant, it might be important to acknowledge the potential relevance of smaller clusters too, such as 1D dimers, and 2D triangles and squares. These smaller clusters could potentially contribute to simulated diffraction patterns and offer opportunities to match them with experimentally

observed ones. Therefore, I recommend expanding the discussion in the manuscript to address the possibility of considering these additional cluster types and their potential impact on the interpretation of experimental data.

2. In general, the Li percolation analysis presented in the manuscript is sound. However, it appears that important kinetic effects related to Li-ion mobility in these compounds have been neglected. Specifically, the significance of local environments in influencing diffusion barriers has not been taken into account. If I understand it right, the assumption made in the analysis is that the lattice is rigid and local environments do not impact the energetics of transition states for Li-ion diffusion. It is crucial to highlight and acknowledge this approximation more explicitly in the manuscript, as it could potentially affect the conclusiveness of the obtained results. By emphasizing this approximation, the authors can provide a clearer understanding of the limitations of their approach and the potential impact it may have on the interpretation of the Li percolation analysis. This would enhance the overall transparency of the research and contribute to a more comprehensive assessment of Li-ion mobility in the investigated compounds.

3. The explanation provided in the manuscript for the enhanced capacity observed in LTMO (and even LTMO-500) is plausible. However, it is worth considering other factors that could potentially influence the observed behavior too. One such aspect is anionic-redox activity. It would be valuable if the authors could address this possibility and provide insights based on the presented data. In addition to the impact on the Li-ion connectivity network, SRO might also affect electron transfer processes in this class of compounds. To address this concern, I suggest that the authors discuss the potential role of anionic-redox activity in relation to the observed capacity enhancement. They could evaluate whether the available data support or rule out the involvement of anionic-redox processes. While the primary focus of the study may be on Li-ion connectivity and SRO, considering the influence, for example, of anionic-redox activity would contribute to a more comprehensive understanding of the observed behavior and provide a more robust interpretation of the results.

Reviewer #3 (Remarks to the Author):

In this work, the authors analyzed local atomic arrangements of short-range ordering by combining aberration-corrected STEM, electron diffraction, and a cluster expansion simulation. It was shown that SRO in cation-disordered rocksalt oxide is directly correlated with Li percolation channels affecting the Li transport behavior. It was further revealed that anion doping and heat treatment of the cathode can alter the SRO of the cathode to optimize the performance of the cathode. The findings of the work are certainly interesting and important for providing fundamental insights for designing cation-disordered rocksalt cathodes with improved electrochemical performance by tuning the SRO parameters. However, there are few points that require clarifications before this paper can be recommended for publication in Nature Communications.

1. Z-contrast STEM is a 2D projection of 3D atomic columns in the cation-disordered rocksalt oxide representing average of each column. Would it be possible to predict 3D atomic structures based on the STEM image? This may enable calculation of the Li percolation from experimentally obtained Z-contrast images by generating corresponding atomic structures.

2. The authors states LTMOF has enhanced local ordering than LTMO by comparing their Z-contrast images. Quantification of the ordering may be helpful to determine the degree of SRO for the two materials.
3. How was the simulated structure validated? Is there an agreement in SRO parameter, for example, of the simulated Z-contrast images for LTMO and LTMOF with the experimentally observed ones?
4. What is the Li percolation number for LTMO-500? Will this be larger than LTMO and LTMOF if the value is estimated by providing an atomic structure that gives similar simulated STEM to the experimentally observed image?
5. It was shown that LTMO-500 has better performance than LTMO and LTMOF because of the well-connected Li network. Would it be possible to propose the ideal SRO that gives maximum Li percolation and potentially the optimum performance?
6. Charge profile of LTMOF shows monotonically increasing voltage which is different from LTMO and LTMO-500 with a reaction plateau after 150 mAh/g. Is the lower performance of LTMOF solely associated with the Li percolation property from the SRO? Does the presence of F cause reduced reaction potential and/or Li diffusivity?

REVIEWER COMMENTS

Reviewer #1 (Remarks to the Author):

The manuscript provides a way to unravel short-range order (SRO) in Li-excess cation-disordered rocksalt (DRO) using aberration-corrected scanning transmission electron microscopy, electron diffraction, and cluster-expansion Monte Carlo simulations. The authors also claim that they can tune the SRO by substitution of transition metals.

Answer: We sincerely thank the reviewer for the positive comment.

I have the following questions:

1) If the SRO in DRX are a convolution of octahedron, tetrahedron and cube type, is this a dynamic or a static ordering?

Answer: Thanks to the reviewer for asking this. The SRO we are referring to is a statistic average of all local atomic configurations in the bulk phase. It is a static local ordering in terms of local structure.

2) How do the atomic-scale STEM Z-contrast images and the fast Fourier transform (FFT) intensity color maps correlate with diffusion properties determined with e.g. GITT or impedance spectroscopy.

Answer: Thanks to the reviewer for this excellent point. The atomic-scale STEM Z-contrast images and FFT are used to resolve the SRO in DRX materials. They do not directly provide diffusivity information, but reveals the atomic-level structural information, which can be correlated to the connectivity of local Li-rich channels. Moreover, we use cluster expansion Monte Carlo simulations to reproduce the atomic structures as seen in experiments. At this sense, the observed atomic-level structural information, especially Li-rich channels, can be correlated to the diffusion properties measured in experiments (e. g. GITT or impedance spectroscopy). A quantitative analysis of such a correlation is not trivial and could be addressed in a dedicated work in future.

3) Why are you using a coating of 70:20:10 and how do you ensure that diffusion properties are not dominated by the carbon/binder matrix?

Answer: We appreciate the reviewer for raising this critical point. The ratio of 70:20:10 (active material:conductive carbon:binder) or similar has been widely used for DRX materials. [Ref: Nature Materials 20.2 (2021): 214-221; Nature 556.7700 (2018): 185-190; Nature communications 7.1 (2016): 13814; Chemical Communications 55.61 (2019): 9027-9030] It has also been demonstrated that electrodes with ratios of 60:30:10, 70:20:10 and 80:15:5 present almost identical discharge voltage profiles confirming that the contribution from carbon/binder matrix is limited.

Reviewer #2 (Remarks to the Author):

This study combines first-principles calculations and experiments to address a significant challenge: the direct atomistic-level visualization of chemical short-range order (SRO) in bulk oxide crystals. Visualizing and, therefore, better understanding SRO is crucial for enhancing the functionality and design of materials across various applications. To tackle this challenge, the authors employ aberration-corrected STEM, electron diffraction, and first-principles simulations, as well as STEM image and electron diffraction simulations. They specifically focus on two Li-excess cation-disordered rocksalt oxides, namely LTMO and LTMOF, as case studies, which is sound. The authors also demonstrate how SRO features can be tailored through anion substitution and thermal treatment, enabling the modification of the electrochemical performance of these materials.

The research findings are presented in a clear and concise manner, and the conclusions drawn seem robust, offering valuable insights for characterizing and designing new electroactive materials. Moreover, experimentalists interested in synthesizing the proposed compounds can benefit from this study. I believe this work holds significant scientific and technological interest, and the methods and approaches described here can be extended to explore other chemical spaces and materials, thus broadening the impact of the research within the field. Overall, the study is comprehensive and well-written. In my opinion, with a few minor revisions as noted below, the manuscript can be published in Nature Communications:

Answer: We would like to thank the reviewer for their appraisal of this work.

1. One aspect that requires clarification is the limited consideration of SRO types restricted to tetrahedron, octahedron, and cube clusters. While these clusters are indeed significant, it might be important to acknowledge the potential relevance of smaller clusters too, such as 1D dimers, and 2D triangles and squares. These smaller clusters could potentially contribute to simulated diffraction patterns and offer opportunities to match them with experimentally observed ones. Therefore, I recommend expanding the discussion in the manuscript to address the possibility of considering these additional cluster types and their potential impact on the interpretation of experimental data.

Answer: Thanks to the reviewer for this excellent point. The reason why we focus on tetrahedron, octahedron and cube clusters is because these are the dominating SRO we have seen in simulation and characterization. So far, we did not observe other SRO features in the widely studied Mn-based DRX compounds. We have clarified this point in page 4 colored in red. The specific context is also attached below.

“It is worth mentioning that our analysis of the three types of SRO features is based on our observation that these features frequently show up during both experiments and simulations. Our

analysis does not eliminate the existence of other possible SRO feature; however, we also did not see other important SRO feature in the material systems we investigated.”

Additionally, we appreciate the reviewer’s suggestion of considering other clusters, such as 1D dimers, 2D triangles and squares etc. These are indeed potential SRO clusters. It is worth to highlight that these are also building pieces of the tetrahedron, octahedron and cube clusters. Our discovery of these three dominating SRO features reveals that even though 1D dimers, 2D triangles and squares may exist, there are additional constraints on these local structures to further assemble them into tetrahedron, octahedron and cube clusters.

To clarify this point, we have added additional description in page 4 colored in red. The specific context is also attached below.

“Moreover, lower dimension ordering within local clusters, such as 1D dimer, 2D triangle or square may also exist, since they are the building blocks of the 3D tetrahedron, octahedron and cube type SRO. Our analysis implies that the lower-dimension local ordering, if existing, has to satisfy the constraint as defined by the 3D SRO structures.”

2. In general, the Li percolation analysis presented in the manuscript is sound. However, it appears that important kinetic effects related to Li-ion mobility in these compounds have been neglected. Specifically, the significance of local environments in influencing diffusion barriers has not been taken into account. If I understand it right, the assumption made in the analysis is that the lattice is rigid and local environments do not impact the energetics of transition states for Li-ion diffusion. It is crucial to highlight and acknowledge this approximation more explicitly in the manuscript, as it could potentially affect the conclusiveness of the obtained results. By emphasizing this approximation, the authors can provide a clearer understanding of the limitations of their approach and the potential impact it may have on the interpretation of the Li percolation analysis. This would enhance the overall transparency of the research and contribute to a more comprehensive assessment of Li-ion mobility in the investigated compounds.

Answer: We appreciate the critical suggestion from the reviewer. The consideration of OTM channel percolation is a simplified framework that consider OTM channel as the dominating local structure that is relevant to Li diffusion (Science 343, 519-522 (2014)). However, we agree with the reviewer that in realistic situations, the Li diffusion can be relevant to many other local structures. Another limitation is that transition metal migration may significantly alter the distribution of OTM channels, this is something we did not consider in this work.

To clarify this point, we have added relevant discussions in Page 9-10 and also attached below.

“The consistence between theoretical prediction and electrochemical performance validates that theoretical framework of OTM percolation is useful for understanding the design handle for promoting Li diffusion. However, it is worth mentioning that such analysis is only based on the cation distribution of fully lithiated DRX structures, while assuming the formation of OTM local structure. In that case, such framework does not consider the impact of other local structures and the possibilities of large-scale rearrangement of cations after cycling.”

3. The explanation provided in the manuscript for the enhanced capacity observed in LTMO (and even LTMO-500) is plausible. However, it is worth considering other factors that could potentially influence the observed behavior too. One such aspect is anionic-redox activity. It would be valuable if the authors could address this possibility and provide insights based on the presented data. In addition to the impact on the Li-ion connectivity network, SRO might also affect electron transfer processes in this class of compounds. To address this concern, I suggest that the authors discuss the potential role of anionic-redox activity in relation to the observed capacity enhancement. They could evaluate whether the available data support or rule out the involvement of anionic-redox processes. While the primary focus of the study may be on Li-ion connectivity and SRO, considering the influence, for example, of anionic-redox activity would contribute to a more comprehensive understanding of the observed behavior and provide a more robust interpretation of the results.

Answer: Thanks to the reviewer for this inspiring point for further discussion. We agree with the reviewer that the anionic-redox process maybe relevant. We do agree that SRO may also affect electron transfer and consequential anionic redox potential. However, we think the impact of distinct anionic-redox will be tiny when comparing LTMO with LTMO-500, especially considering these two materials have the exact same composition. In DRX, cation-disordering allows the formation of a significant amount of linear “Li-O-Li” configuration which produces states that can be oxidized [Ref: Nat. Chem., 2016, 8, 692–697], and is therefore regarded as the key structural origin of oxygen redox. With a high Li-excess content of 20% in the model materials, there should already be more than enough “activated” O centers with Li-O-Li configuration. Moreover, several of our previous studies independently demonstrated that the Li transport kinetics, instead of the amount of (anionic) redox reservoir, is the critical control of the accessible capacity of DRX materials [Adv. Energy Mater. 2020, 10, 1903240, Nature Materials 20.2 (2021): 214-221; Advanced Energy Materials 12.21 (2022): 2103923; Nature Energy 5.3 (2020): 213-221.].

To clarify this point, we added the following text. “*It should be noted that SRO may also affect electron transfer and consequentially anionic redox potential. In DRX, cation-disordering allows the formation of a significant amount of linear “Li-O-Li” configuration which produces states that can be oxidized [Ref: Nat. Chem., 2016, 8, 692–697], and is therefore regarded as the key structural origin of oxygen redox. With a high Li-excess content of 20% in the model materials, there should already be more than enough “activated” O centers with Li-O-Li configuration. Moreover, it has been demonstrated that the Li transport kinetics, instead of the amount of (anionic) redox reservoir, is the critical control of the accessible capacity of DRX materials [29Adv. Energy Mater. 2020, 10, 1903240, Nature Materials 20.2 (2021): 214-221; 7Advanced Energy Materials 12.21 (2022): 2103923; Nature Energy 5.3 (2020): 213-221.]. It would be expected that, as comparing LTMO with LTMO-500, the anionic-redox will not show significant difference, especially considering these two materials have the exact same composition.*”

Reviewer #3 (Remarks to the Author):

In this work, the authors analyzed local atomic arrangements of short-range ordering by combining aberration-corrected STEM, electron diffraction, and a cluster expansion simulation. It was shown that SRO in cation-disordered rocksalt oxide is directly correlated with Li percolation channels affecting the Li transport behavior. It was further revealed that anion doping and heat treatment of the cathode can alter the SRO of the cathode to optimize the performance of the cathode. The findings of the work are certainly interesting and important for providing fundamental insights for designing cation-disordered rocksalt cathodes with improved electrochemical performance by tuning the SRO parameters. However, there are few points that require clarifications before this paper can be recommended for publication in Nature Communications.

Answer: We sincerely thank the reviewer for the highly positive comment to our work.

1. Z-contrast STEM is a 2D projection of 3D atomic columns in the cation-disordered rocksalt oxide representing average of each column. Would it be possible to predict 3D atomic structures based on the STEM image? This may enable calculation of the Li percolation from experimentally obtained Z-contrast images by generating corresponding atomic structures.

Answer: Thanks to the reviewer for this excellent question. Regular Z-contrast STEM imaging only measures the statistic average in each atomic column, while the distribution of ions in the atomic column is unknown. In order to reconstruct the 3D structure, more advanced methods, such as atomic electron tomography (AET) (Nature, 592, 60–64 (2021)) are required. The AET method requires a nano-sized sample to be imaged by a regular STEM many times (> 50) at the same nanoscale location, at a series of tilting angles, to perform the 3D reconstruction. This requires the sample to be very robust under electron beam, *i.e.*, damage-free after extensive electron dose. With the current technical limitations, the battery materials are generally not suitable for such an imaging. We totally agree with the reviewer that would be future work along with the low dose and sensitive detector techniques.

2. The authors states LTMOF has enhanced local ordering than LTMO by comparing their Z-contrast images. Quantification of the ordering may be helpful to determine the degree of SRO for the two materials.

Answer: We thank the reviewer for this excellent point. As being indicated in reply to question 1, it is a very challenge to reproduce exact 3D atomic structure with TEM characterization. Therefore, we rely on cluster expansion Monte Carlo (CEMC) to leverage our understanding of characterizations. We first apply CEMC simulation to reproduce the same diffraction pattern as seen in experiment, in which case we have the simulated 3D atomic structures. Quantitative analysis can then be done on these simulated atomic structural models. Fig. 3 is an example of the quantitative ordering analysis we can did by comparing LTMOF with LTMO.

3. How was the simulated structure validated? Is there an agreement in SRO parameter, for

example, of the simulated Z-contrast images for LTMO and LTMOF with the experimentally observed ones?

Answer: That is an excellent point. Given that the SRO parameter cannot be directly measured experimentally. We verify the simulated structure by comparing the diffusive scattering features from both simulation and experiments. As being shown by Fig. 1 (a), (b) in comparison with Fig. 1(i), (j), our simulation can reproduce the important features (e. g. ring like feature and square like feature) as seen in experiment. In that case, we can rely on our simulated structures to provide more insight on Li percolation and local structure evolution.

4. What is the Li percolation number for LTMO-500? Will this be larger than LTMO and LTMOF if the value is estimated by providing an atomic structure that gives similar simulated STEM to the experimentally observed image?

Answer: Thanks to the reviewer for this great point. It is difficult to directly simulate LTMO-500, as LTMO-500 is generated with annealing at 500 °C for 60h. It is not possible to directly simulate the annealing process for 60h. Therefore, we cannot answer this question from a simulation perspective. However, we would say that the Li-percolation number for LTMO-500 should be larger than LTMO and LTMOF, based on our STEM observations and electrochemical measurements.

5. It was shown that LTMO-500 has better performance than LTMO and LTMOF because of the well-connected Li network. Would it be possible to propose the ideal SRO that gives maximum Li percolation and potentially the optimum performance?

Answer: Thanks to the reviewer for this insightful question. If one can enumerate all possible SRO, it is possible to propose the ideal SRO that gives maximum Li percolation and potentially optimum performance. In this work, we limited our scope to only Mn-Ti based DRX, and the exploration of SRO types and features are only based on our experimental observations of the SRO in LTMO and LTMOF. The open question of “which SRO is best” actually suggests that there are still many unresolved questions in the field, which will attract numerous further studies.

6. Charge profile of LTMOF shows monotonically increasing voltage which is different from LTMO and LTMO-500 with a reaction plateau after 150 mAh/g. Is the lower performance of LTMOF solely associated with the Li percolation property from the SRO? Does the presence of F cause reduced reaction potential and/or Li diffusivity?

Answer: Thanks to the reviewer for this critical point. We want to clarify that the elevation of voltage curve is a consequence of increased capacity under same voltage window, which should not be over-interpreted. Moreover, the shape of the voltage profile of DRX cathode is determined by various factors, such as Li-site energy distribution (related to SRO), metal redox potential, tetrahedral Li formation etc. (Chem. Mater. 2016, 28, 15, 5373–5383). Fluorination does not necessarily reduce the reaction potential, instead it strongly modifies the SRO which could indirectly change the voltage profile. A more detailed description of such convoluted effect is elaborated in Adv. Energy Mater. 2020, 10, 1903240.

REVIEWERS' COMMENTS

Reviewer #3 (Remarks to the Author):

The authors have responded to reviewers' comments well and made revisions to the manuscript, which resolved the questions and concerns raised by the reviewers. It is understandable that it is not easy to perform some of the suggested experiments and analyses to resolve all of the questions with this manuscript, but the results shown in this work will certainly attract attentions and stimulated further research in the related fields. The revised manuscript is now recommended for publication in Nature Communications.

RESPONSE to REVIEWER COMMENTS

Reviewer #3 (Remarks to the Author):

The authors have responded to reviewers' comments well and made revisions to the manuscript, which resolved the questions and concerns raised by the reviewers. It is understandable that it is not easy to perform some of the suggested experiments and analyses to resolve all of the questions with this manuscript, but the results shown in this work will certainly attract attentions and stimulated further research in the related fields. The revised manuscript is now recommended for publication in Nature Communications.

Answer: We sincerely thank the reviewer for the positive comment.